# Effects of Drought and Flooding on Phytohormones and Abscisic Acid Gene Expression in Kiwifruit

**DOI:** 10.3390/ijms24087580

**Published:** 2023-04-20

**Authors:** Kirstin V. Wurms, Tony Reglinski, Poppy Buissink, Annette Ah Chee, Christina Fehlmann, Stella McDonald, Janine Cooney, Dwayne Jensen, Duncan Hedderley, Catherine McKenzie, Erik H. A. Rikkerink

**Affiliations:** 1Ruakura Research Centre, The New Zealand Institute for Plant and Food Research Limited, Hamilton 3214, New Zealand; 2Mount Albert Research Centre, The New Zealand Institute for Plant and Food Research Limited, Auckland 1025, New Zealand; 3Palmerston North Research Centre, The New Zealand Institute for Plant and Food Research Limited, Palmerston North 4410, New Zealand; 4Te Puke Research Centre, The New Zealand Institute for Plant and Food Research Limited, Te Puke 3182, New Zealand

**Keywords:** ABA catabolism, ABA synthesis, abiotic stress, *Actinidia*, biomarkers, hormonal crosstalk, molecular markers, plant hormones, water deficit, waterlogging, water stress

## Abstract

Environmental extremes, such as drought and flooding, are becoming more common with global warming, resulting in significant crop losses. Understanding the mechanisms underlying the plant water stress response, regulated by the abscisic acid (ABA) pathway, is crucial to building resilience to climate change. Potted kiwifruit plants (two cultivars) were exposed to contrasting watering regimes (water logging and no water). Root and leaf tissues were sampled during the experiments to measure phytohormone levels and expression of ABA pathway genes. ABA increased significantly under drought conditions compared with the control and waterlogged plants. ABA-related gene responses were significantly greater in roots than leaves. ABA responsive genes, *DREB2* and *WRKY40*, showed the greatest upregulation in roots with flooding, and the ABA biosynthesis gene, *NCED3*, with drought. Two ABA-catabolic genes, *CYP707A i* and *ii* were able to differentiate the water stress responses, with upregulation in flooding and downregulation in drought. This study has identified molecular markers and shown that water stress extremes induced strong phytohormone/ABA gene responses in the roots, which are the key site of water stress perception, supporting the theory kiwifruit plants regulate ABA to combat water stress.

## 1. Introduction

Global warming is predicted to increase the frequency and intensity of extreme weather events such as drought and flooding. Detrimental environmental conditions can cause reductions in plant growth, affecting yield quality and quantity, and can result in significant crop losses [1,2,3,4,5,6]. Water stress, caused by drought and waterlogging, is the most significant constraint to crop production worldwide today [2,4,6] and research to interrogate the associated stress-response mechanisms is essential to facilitate the development of climate resilient varieties [2,3,4]. Drought stress occurs in plants when there is low water availability for a prolonged period [2,3], which adversely affects nutrient uptake and translocation, and disrupts gas exchange and carbon assimilation. This together with increased oxidative stress results in reduced growth and yields and may even lead to plant death [2,3].

Flooding can cause waterlogging where the roots are saturated and/or submergence, where parts of the plant above ground are partially or completely covered with water [1,6]. Waterlogging aids pathogen infection, reduces oxygen availability and restricts gas exchange in submerged tissues causing an accumulation of toxic end-products [1]. Submergence can also increase pathogen infection, osmotic and salt stress, and reduces transpiration, nutrient uptake and transport leading to energy-starved tissues. Prolonged submergence, however, results in the death of most terrestrial plants [1]. 

This study focused on kiwifruit as the exemplar crop for water stress studies because the kiwifruit industry makes a considerable contribution to the New Zealand economy, with kiwifruit being the top horticultural export [7,8]. Although previous work has made significant progress in elucidating complex phytohormone and gene expression networks involved in plant response to water stress, little has been done in kiwifruit, especially in terms of the flooding response. The response to water stress was examined in the two most economically important cultivars in New Zealand, the green-fleshed kiwifruit cultivar *Actinidia chinensis* var. *deliciosa* ‘Hayward’ and the yellow-fleshed cultivar *A. chinensis* var. *chinensis* ‘Zesy002’, also known as Gold3 [9]. 

The phytohormone abscisic acid (ABA) has a critical role in regulating the plant response to water stress [2,6,10]. Intracellular concentrations of ABA increase in response to drought stress [4,11] and result in a decrease in stomatal aperture size to reduce water loss, a reduction in photosynthetic rates (due to decreased CO_2_ supply), and the accumulation of protective proteins and metabolites to maintain cellular water status and protect proteins/enzymes/organelles from injury [2,4,12,13]. Conversely, in response to flooding, ABA levels reduce in the submerged tissues (i.e., the roots) of the plant, whilst increasing in the shoots [12,14]. The role of ABA in flooding tolerance is less well understood than in drought [6]. Moreover, ABA metabolism appears to exhibit tissue specific responses in the leaves and roots, depending on the plant species and water stress duration [12,13,15]. 

Endogenous ABA levels are dynamically controlled through regulation of ABA biosynthesis, catabolism/storage (both catalytic hydroxylation and ABA conjugation to create storage forms of ABA) and transportation [2,12,16,17]. The roots are the site where water stress signals are first perceived and ABA biosynthesis occurs, with ABA being transported through the vasculature to the shoot once formed [2]. Oxidation is the primary pathway for ABA catabolism [17,18] and is a mechanism for ABA inactivation [17]. Catabolic products are also capable of activating a subset of ABA receptors, allowing further plant responses to long-term stress conditions [12]. ABA conjugation, principally with glucose, that is carried out by *ABA glucose ester*, is also important because it creates an inert storage form of ABA, which can be readily re-activated (without the need for de-novo synthesis) by deglycosylation in the vacuole by *beta 1–3 glucosidase* [16,19]. 

While ABA is the key phytohormone involved in regulating plant responses to stress (especially abiotic stress), it also interacts with other phytohormones through crosstalk in signal pathways to regulate many developmental processes and the general stress response [1,5,10,20]. Changes in endogenous concentrations of phytohormones such as salicylic acid (SA) and jasmonic acid (JA) are more commonly associated with defence regulation but are also affected by crosstalk with ABA [20,21,22]. Thus, the concentrations and ratios of these phytohormones should be considered when investigating the plant response to water stress. Consequently, the concentrations and ratios of ABA, SA, JA, along with some of their key pre-cursors, conjugates and catabolites were also monitored in this study.

The aims of this research were to investigate the molecular basis of the kiwifruit response to water stress and to identify possible marker genes across the ABA signalling network in kiwifruit. We also investigated if there were transcriptome signals that could differentiate between the two water stress extremes analysed. Genes representing all these areas were examined in the current study and were selected based on their role as key marker genes in other studies [2,4,9,11,14,16,20,23,24]. Results showed that of all the phytohormones measured, endogenous ABA concentrations increased consistently in drought conditions, especially in the roots, and were associated with upregulation of ABA synthesis genes. Conversely, decreased ABA concentrations in waterlogged roots were associated with increased activity of ABA catabolic genes. This supports the theory that the ABA pathway plays a key role in kiwifruit response to water stress. Understanding these molecular responses and how to manipulate them will help to increase kiwifruit resilience and productivity during water stress conditions, which is becoming more crucial as the pace of climate change accelerates [4,5]. 

## 2. Results

### 2.1. Soil Moisture Content Increases with Flooding and Decreases with Drought 

Relative percent changes in pot weights over the course of all four trials relative to the control treatment (Table 1) showed that moisture content decreased significantly in the drought treatment for both kiwifruit cultivars (up to a 30% decrease, *p* < 0.001) and increased significantly in the flooding treatment (up to a 17 % increase, *p* < 0.001). Since the soil constitutes most of the weight in each pot, the weight changes are indicative of significant changes in soil moisture content. 

### 2.2. Water Stress Affects Concentrations of Phytohormones and Their Derivatives in a Tissue Specific Manner

Experiments were performed to compare changes in phytohormone concentrations of SA, JA, ABA, indole acetic acid (IAA) and their derivatives over time in potted kiwifruit plants grown under different watering regimes; regular watering (control), unwatered, and waterlogged (i.e., the lower third of each pot was immersed continuously in water). A preliminary study showed that leaves on ‘Zesy002’ potted plants reached their wilt point after 48–72 h without water and, therefore, leaf and root tissue was collected before and after this period in four subsequent experiments. Only significant results (Appendix A) that were consistent across at least two experiments are presented here. 

ABA and the ABA-related metabolites dihydrophaseic acid (DPA) and phaseic acid (PA) showed utility as water stress markers (Figure 1, Figure 2 and Figure 3), with concentrations of ABA increasing in the leaves with both types of water stress and in the roots with drought across all four experiments (Figure 1). The effects of waterlogging on ABA concentration in the roots were more variable, with no effect in the 2020 experiments on ‘Zesy002’ and a significant decrease in both ‘Zesy002’ and ‘Hayward’ roots in 2021 (Figure 1). Overall, the largest increases in ABA concentration (up to 20-fold) were observed in unwatered roots, although total ABA concentration is 6- to 10-fold lower in roots than in leaves for both cultivars (Figure 1). For DPA, concentrations increased significantly in the roots in unwatered treatment (up to 7-fold in unwatered ‘Hayward’ roots in 2021) (Figure 2). 

Other results were less consistent, but an increase of up to 4-fold in DPA was seen in ‘Hayward’ leaves in 2021 and much smaller increases in DPA concentrations in the leaves (up to 2-fold) in the unwatered ‘Zesy002’ leaves (Figure 2). The most consistent effect for PA was a significant increase (up to 5-fold) in unwatered roots (Figure 3). PA concentrations in ‘Hayward’ leaves increased by 3-fold after 96 h in unwatered and waterlogged plants compared with controls but not in ‘Zesy002’ in 2021 (Figure 3). However, it should be noted that basal concentrations of PA were 2x higher in ‘Zesy002’ leaves (646 ng/gFwt) than ‘Hayward’ (297 ng/gFwt).

Other phytohormones that showed significant treatment effects in root tissue in two or more experiments included 7-hydroxy-abscisic acid (7-OH-ABA), a minor catabolite of ABA, which decreased with restricted water availability in the 2020 trials (Appendix A). Jasmonic acid or JA (Appendix A) and its precursor 3-oxo-2-(2-pentenyl)-cyclopentane-1-butyric acid or OPC-4 (Appendix A) decreased with waterlogging. 9,10-Dihydrojasmonic acid (DH-JA), a less active form of JA (Appendix A), and salicylic acid O-β-glucoside (SAG), a storage form of SA (Appendix A), increased with waterlogging in roots. Abscisic acid glucosyl ester (ABA-GE), a storage form of ABA (Appendix A), increased with both drought and waterlogging. In leaves, the effects of water stress on other phytohormones were generally inconclusive, apart from 7-OH-ABA, which increased with both types of water stress in the 2020 experiments (Appendix A).

The experiments in 2021 compared hormonal response to water stress in ‘Zesy002’ and ‘Hayward’ kiwifruit. The hormonal response patterns indicated a similarity between the cultivars with respect to main treatment effects (Appendix A). The main cultivar specific change was that the average concentration of the auxin indole-3-acetic acid (IAA) in ‘Zesy002’ roots was significantly higher (*p* = 0.000) in waterlogged plants than in the other treatments (Appendix A), whereas in ‘Hayward’ IAA was significantly lower in waterlogged and unwatered plants than in the controls. 

### 2.3. Gene Expresison Depends on the Type of Water Stress and the Tissue Sampled

Gene expression data are presented as heat maps (Figure 4, Figure 5 and Figure 6), which show the fold change in gene count data (log_2_ transformed for statistical analysis, then back transformed) relative to the control treatment at each sampling time point for three experiments, one of which was carried out in 2020 (Figure 4) and two in 2021 (Figure 5 and Figure 6). Although 18 genes of interest (GoI) were measured by NanoString, only those genes showing statistically significant increases or decreases in expression (as indicated by asterisks on the heat maps) are presented here and only expression patterns that were consistent across two or more experiments are highlighted below. In all three experiments, the average counts of *ABA-insensitive 4-i* and *-ii* (*ABI4-i* and -*ii*, involved in ABA signal transduction), *Myelocytomatosis 2-like* (*MYC2-like*, an ABA-responsive transcription factor (TF) involved in JA/ethylene (ET) pathway crosstalk), and *Responsive to desiccation 29b* (*RD29B*, an ABA-dependent drought-responsive gene) were only just above the background noise levels as indicated by the negative controls included in every sample.

A key trend consistent across all experiments was that more genes tended to be upregulated to a greater extent (larger fold-changes) in the roots than in the leaves with water stress (Figure 4, Figure 5 and Figure 6). This was particularly the case for *DRE-binding protein 2* (*DREB2*, an ABA-independent gene that induces expression of proteins that increase tolerance to abiotic stress) *cytochrome P450-type monooxygenases-i and -ii* (*CYP707A-i* and *ii*, ABA catabolic enzymes), *nine-cis-epoxycarotenoid dioxygenase 3* (*NCED3*, the key regulatory enzyme in ABA synthesis) and *WRKY-40* (an ABA-responsive TF involved in SA/ABA crosstalk). In contrast, *CYP707A-iii* was more strongly upregulated in the leaves than in the roots, especially in the unwatered treatment, where upregulation was higher than in the waterlogged treatment. In addition to showing the most consistent expression, *DREB2*, the three *CYP707A* isoforms, *NCED3* and *WRKY-40* were the most strongly upregulated of all 18 GoI (Figure 4, Figure 5 and Figure 6).

Several genes were able to distinguish between the water stress treatments, particularly in the roots. These were *CYP707A-i* and *ii*, which were downregulated in the roots in drought-like conditions but upregulated during waterlogging, with both responses appearing quite durable, lasting up to 112 h in the unwatered treatment and up to 48 h in waterlogged roots (Figure 4, Figure 5 and Figure 6). In addition, *WRKY-40* was strongly upregulated by waterlogging but not by drought in the roots and showed the highest fold changes of any GoI in both cultivars in 2021 (78.3-fold for ‘Zesy002’ and 104.5-fold for ‘Hayward’, Figure 5 and Figure 6). 

*DREB2* was the second most highly upregulated gene in waterlogged roots in 2021, with maximum expression occurring at 48 h (Figure 5 and Figure 6) and was the most highly upregulated gene in the same treatment and tissue in 2020 (Figure 4). Upregulation persisted for the duration of the sampling period (Figure 4, Figure 5 and Figure 6). *DREB2* was also upregulated in the unwatered treatment in the roots, but expression peaked later at 96–112 h and less strongly than with waterlogging (up to 30-fold less). Upregulation in the unwatered treatment in roots also tended to be for the duration of experiment. The unwatered treatment was also associated with a less extreme *DREB2* upregulation in the leaves, especially in ‘Hayward’ (up to 11-fold, Figure 6). 

*NCED3* was most strongly upregulated by drought in the unwatered treatment in both leaves (up to 15.3-fold, Figure 4, Figure 5 and Figure 6) and roots (up to 18.8-fold, Figure 4, Figure 5 and Figure 6). Upregulation also occurred in the waterlogged treatment in both leaves (up to 5.8-fold) and roots (up to 7.5-fold), but it was weaker than the drought response (Figure 4, Figure 5 and Figure 6). 

Significant differential expression of *Pathogenesis-related protein family 1* (*PR1* a salicylic acid (SA) pathway marker, responsive to biotic stress) was observed in both ‘Zesy002’ experiments (Figure 4 and Figure 5), but not in the ‘Hayward’ trial (Figure 6). In the ‘Zesy002’ trials, *PR1* was upregulated in the unwatered treatment in the leaves and the roots, but expression patterns in the waterlogged treatment for both tissue types were inconsistent (Figure 4 and Figure 5).

Both types of water stress resulted in *Responsive to desiccation 22* (*RD22*) upregulation in the leaves, but expression patterns in the roots were more variable (Figure 4, Figure 5 and Figure 6). Although the average counts of *RD29B* were just above the limits of detection, the unwatered treatment was associated with significant upregulation in the leaves and more variable expression with waterlogging in both tissue types (Figure 4, Figure 5 and Figure 6). 

*WRKY-40* was strongly upregulated in the roots in the unwatered treatment for the duration of all three trials (Figure 4, Figure 5 and Figure 6). A comparison of Figure 5 and Figure 6 shows that overall expression patterns are similar for both cultivars. 

The remaining genes shown in the heat maps are not specifically mentioned because either consistent treatment patterns were not seen in at least two experiments, or levels of expression rarely changed greater than 0.5–2 fold, which is considered by many to be statistically unreliable [25,26,27], and/or expression fluctuated continuously between upregulation and down regulation across the sampling time points. 

## 3. Discussion

Phytohormones modulate the plant response to biotic and abiotic stress [22]. In this study the levels of major phytohormones (ABA, SA, JA, IAA, and associated derivatives) were compared in kiwifruit plants experiencing water deficit or flooding. Only ABA and its catabolic derivatives DPA and PA increased significantly (up to 20-fold for ABA and 5- to 7-fold for the derivates) and consistently in drought conditions relative to the control, especially in the roots, whilst responses to flooding were more variable. This concurs with the known role of ABA in modulating drought stress in other plant species [2,14,28]. Endogenous levels of ABA are regulated through a balance of synthesis, catabolism, storage and transport [12] and several gene candidates involved with ABA homeostasis in kiwifruit were identified in the water stress experiments. *DREB2* (an ABA-independent drought response gene) and *WRKY40* (an ABA-responsive gene) showed the greatest upregulation in roots with flooding, whereas *NCED3* (an ABA biosynthesis gene) was most strongly upregulated in the roots during drought. Members of the CYP707A family (which catabolize ABA to inactive PA), responded differentially to water stress, with *CYP707A i* and *ii* upregulated by flooding but downregulated in drought in the roots, whilst *CYP707A iii* showed the greatest upregulation with drought in the leaves. This suggests that kiwifruit may respond to drought by increasing ABA concentrations in both leaves and roots to restrict growth and thereby increase the root:shoot ratio and promote the growth of lateral roots [12]. Conversely, decreased ABA concentrations in waterlogged roots will promote rapid shoot extension enabling the plant to grow above the shallower depth flooding that is typical of waterlogging [6]. Whilst patterns of ABA-related gene expression were tissue specific and dependent on the type of water stress (drought versus flooding), there were no marked differences in water stress induced gene expression between the ‘Hayward’ and ‘Zesy002’ kiwifruit cultivars. Similar results, particularly regarding the expression of *NCED3* and *CYP707A*, were observed in *Arabidopsis*, citrus, and rice under water stress [2,14,18]. The observed phytohormone/ABA gene responses in the roots, which are the main site of water stress perception, suggest the importance of ABA in the water stress response of kiwifruit. To the best of our knowledge, this represents the first published study looking at diverse hormone and gene responses to different water stress conditions in kiwifruit. 

### 3.1. The Drought Response

Drought is the most important climate factor impacting agricultural productivity, and the frequency and severity of drought events is increasing due to climate change [4,23]. Numerous studies have established that ABA is the key phytohormone involved in the drought response and plant tolerance and/or resistance to this type of stress [2,11,12,23]. Plant cells typically accumulate ABA in response to dehydration, especially in roots, which are the site where water stress is first perceived [2,4]. In the current study, ABA concentrations were consistently most elevated (up to 20-fold) in kiwifruit roots under drought conditions, although ABA concentration increased in the leaves to a lesser extent (up to 14-fold). Total ABA concentrations were higher in the leaves than the roots (both pre- and post-stress) since ABA synthesis and accumulation in guard cells in leaves plays an important role in limiting water loss through transpiration by controlling stomatal closure [2]. Moreover, expression of the ABA-synthesis marker, *NCED3*, was upregulated up to 18.8-fold in roots and 15.3-fold in leaves of unwatered ‘Zesy002’ plants. In *Arabidopsis*, *NCED3* is a dominant contributor to ABA synthesis under drought stress by acting as the key regulatory enzyme [2,11,17]. Similarly, an increase in endogenous ABA was associated with upregulation of *NCED3* and reduced *CYP707A* expression in citrus roots under drought [14]. The phylogenetic evidence in favor of the kiwifruit candidate genes chosen for *NCED3* and *CYP707A* performing the same function as their *Arabidopsis* counterparts was convincing, as they grouped with candidate genes from other species as expected based on the evolutionary relationships between the species present in their EnsemblPlants trees (Appendix A). Remnants of conserved gene order (micro-synteny) were also identified in both cases (either with the *Arabidopsis* gene or with other Rosids), providing further evidence that these genes and some of their neighboring genes are related by descent. Given that *Arabidopsis* is an annual herbaceous weed, whilst citrus and kiwifruit are woody perennials from the Rosid and Asterid dicot superclades, respectively, this is indicative of a highly conserved response in eudicots that is likely to be key in overcoming drought stress. Furthermore, experiments with transgenic lines of two monocots, wheat and rice, showed that increased drought tolerance was associated with elevated ABA sensitivity, resulting in greater endogenous ABA accumulation [29,30]. This suggests that the role of ABA in drought tolerance may be conserved across angiosperms as a whole.

The concentration of ABA catabolites (DPA and PA) was also greatest (5- to 7-fold higher) in water deficient kiwifruit roots indicating an upregulation of ABA metabolism (both biosynthesis and catabolism) as a whole. Increases in concentrations of ABA, DPA and PA were also observed in *Arabidopsis* plants after the onset of dehydration [20]. 7-OH-ABA, a minor ABA catabolite, decreased in unwatered ‘Zesy002’ roots in the 2020 experiments. Hydroxylation of ABA to 7-, 8- or 9-OH-ABA precedes the isomerisation to the inactive products PA and DPA, but the main route of catabolism is usually via 8-OH-ABA [16].

Accumulation of ABA in tissues functions to offset drought in three ways–controlling stomatal closure (a rapid response), via antagonistic interactions with the GA pathway which affects growth (a slower response), and by downstream signalling effects (a slower response), for example induced accumulation of protectants that confer stress tolerance by regulating redox balance or modifying ion transport to re-establish homeostasis [2,12]. Expression of *RD22* and *RD29B* are examples of ABA-dependent genes involved in downstream signalling effects and these increased in response to drought consistently in kiwifruit leaves, although responses in roots were variable. Similar increases of *RD29B* expression have been observed in *Arabidopsis* [20]. 

Crosstalk between ABA and other phytohormone pathways can also play a role in regulating the drought response but are more complex to unravel. In the current study, IAA decreased in unwatered ‘Hayward’ roots but not in ‘Zesy002’. Further investigation is required to determine if this is a cultivar difference. A cultivar-specific response was also observed in IAA levels in waterlogged roots.

*PR1* is a classic marker for the SA pathway in plants, and has been shown to be induced by biotic stress and by a SA-elicitor in ‘Zesy002’ vs. ‘Hayward’ leaves [9,24,31]. In this study, SA content increased in ‘Zesy002’ leaves with water stress, but not in ‘Hayward’, suggesting that ‘Zesy002’responds more acutely to abiotic stress via the SA pathway than ‘Hayward’. Moreover, this also indicates potential for *PR1* to function as a broader stress-response marker. There also is evidence from other plant species which suggests that SA plays an important role in water stress (mainly drought), e.g., foliar application of SA has been used to mitigate water stress in tomato, spinach and pistachio [3,32]. Although the SA and ABA pathways are usually antagonistic, this is not always the case [33]. 

Studies of *Arabidopsis* mutants have revealed that plant responses to drought stress involve both ABA-dependent and -independent signaling pathways, with ABA-independent pathways regulated by the DREB1 and DREB2 gene families [11,12,23]. *DREB2* also appears to play a role in the ABA-independent response of kiwifruit to drought stress, since it was upregulated up to 6.8-fold in ‘Zesy002’ roots and up to 5.2-fold in ‘Hayward’ but, surprisingly, increases were even greater with flooding (up to 74.7-fold). The involvement of DREBs in water stress makes sense as they belong to a major subgroup of the Apetala 2 (AP2) transcription factors and coordinate multiple stress responses at the transcriptional level (often together with basic leucine zipper or bZIP proteins) [12,23].

### 3.2. Response to Flooding

Flooding (including both waterlogging and submergence) is the second most damaging climate factor affecting growth and yield after drought The kiwifruit plants in this research project were subjected to waterlogging. Flooding limits gas exchange in soils resulting in as much as a 10,000-fold reduction in gas diffusion [34]. Consequently, oxygen in the soil rapidly decreases and the soil can become hypoxic (deficient in oxygen) or anoxic (no oxygen) within just a few hours [6]. 

Ethylene (ET) is the main phytohormone governing the flooding response, so the role of ABA in flooding has been studied less, but ABA is also important because of the antagonistic crosstalk between ET/ABA and ABA/gibberellic acid (GA) pathways [1,6,17,18,35]. Flooding induces ABA, but ET being a gas is unable to escape and rapidly builds up in flooded tissues and negatively regulates the concentration of ABA [6]. In *Arabidopsis*, rice and citrus plants, ET negatively regulated the amount of ABA via increased ABA catabolism, (likely mediated by *CYP707A*s), and decreased ABA synthesis by *NCED3*. The net result was an increase of ABA concentrations in the leaves and a decrease in the roots [12,14,18]. Accordingly in the current study, waterlogging resulted in significant increases in *CYP707A-i* and *ii* expression and downregulation of *NCED3*in kiwifruit roots, as well as decreased ABA concentration in the 2021 experiments. ABA concentrations also increased in the leaves. In *Arabidopsis* the *CYP707A* response is spatially, temporally and tissue specific, with each member of the CYP707A gene family playing a slightly different physiological or developmental role [12]. This also appears to be the case in kiwifruit, as *CYP707A-i* and *ii* (most closely related of the three candidates and derived from the most recent *Actinidia* whole genome duplication) were most strongly upregulated in roots of plants in the waterlogging treatment, while *CYP707A-iii* (the most distinct of the three candidates, see Appendix A) was most strongly upregulated in the leaves. It is likely these adaptations in the responsiveness of different CYP707A genes have evolved independently within *Arabidopsis* and kiwifruit as all three kiwifruit candidates evolved since their separation from CYP707A genes in *Arabidopsis* and other Asterid species represented in EnsemblPlants. Endogenous concentrations of phytohormones are determined by the balance between biosynthesis, catabolism, and storage. There are two main pathways for reducing endogenous ABA concentrations–catabolism and storage. Catabolism consists of oxidative breakdown by *CYP707A*s leading eventually to form the catabolic byproducts PA and DPA. Sugar conjugation by glucosyl transferases such *UDP-glycolsyltransferase 71C5* (*UTC-71C5*) form ABA-GE inert storage or transportation forms of ABA [12,16]. The kiwifruit results suggest oxidation was the primary method of reducing ABA because the levels of ABA-GE in the roots were below the level of detection in the 2020 experiments and concentration did not change more than 2-fold with both types of water stress in 2021, and there was no differential expression under water stress of *UTC-71C5*, or of *β-glucanase* that hydrolyzes ABA-GE to ABA. Note, however, that the phylogenetic evidence identifying the kiwifruit functional orthologue for UTC-71C5 is less compelling as this is such a rapidly radiating and evolving plant family (Appendix A). In contrast, there were significant changes in concentrations of up to 3-fold for PA and up to 5-fold for DPA under both types of water stress in leaves and roots, as well a significant increase in expression of all three *CYP707A* catabolic genes. Flooding has also been shown to increase concentrations of DPA in citrus [14]. 

Complex crosstalk between major phytohormone pathways enables tailored responses to different stresses. Concentrations of JA and OPC-4 (a JA precursor) decreased in roots with waterlogging whilst SAG (an inert storage form of SA) and DHJA (a JA catabolite) increased. This suggests possible involvement of the JA ad SA pathways in the kiwifruit response to water stress. Both pathways have been implicated in the drought response [3,21,32], and further analyses of ABA/JA/SA interactions in kiwifruit under more protracted water stress conditions are warranted. 

Auxin (IAA) increased in roots with waterlogging in ‘Zesy002’ and decreased in ‘Hayward’, but more trial work is recommended to determine whether this is a consistent cultivar difference. Cisse et al. [34] found that application of exogenous ABA and IAA promoted waterlogging tolerance in woody myrtaceous plants, suggesting the importance of auxins in the water stress response. 

Whilst *DREB2* (an ABA-independent response gene) is typically thought of as a drought responsive gene, the highest increases in *DREB2* expression were measured in waterlogged kiwifruit roots (up to 66.6-fold in ‘Zesy002’ and 74.7-fold in ‘Hayward’). This suggests that *DREB2* plays a more generalized role in response to abiotic stress response than previously thought. This is supported by the findings of Sakuma et al. [36] who demonstrated that overexpression of *DREB2A* in *Arabidopsis* induced not only drought responsive genes, but also salt and heat shock associated genes. 

Interestingly, the ABA-dependent TF *WRKY-40* is also primarily thought of as a drought response gene that represses the expression of *DREB2A* in *Arabidopsis* [23]. However, in kiwifruit *WRKY-40* was strongly upregulated in waterlogged roots (up to 78.3-fold in ‘Zesy002’ and 104.5-fold in ‘Hayward’) and did not adversely affect *DREB2* expression. It is important to note here that DREB2A and DREB2B clades have evolved specifically in the Brassicaceae and therefore the *DREB2* genes in other species are more closely related to the DREB2E and DREB2C clades in *Arabidopsis*. DREB2 expression in other non-Brassicaceae plants under waterlogging conditions merits further investigation to see if the response is akin to that observed in kiwifruit.

### 3.3. Future Directions and Impact

The two main commercial kiwifruit cultivars used in this study were bred for optimal yield production rather than abiotic stress tolerance. However, the *Actinidia* genus is extremely genotypically and phenotypically diverse [37] and there is likely potential to mine the germplasm for water stress tolerance characteristics. 

This research focused on responses during the onset water stress and did not examine mechanisms associated with recovery after removal of the stress. This knowledge gap should be addressed to better understand water stress resilience in kiwifruit. For instance, in citrus roots the restoration of control levels of ABA after flooding has been associated with increased levels of *β-glucanase* activity which hydrolyses ABA-GE [14].

A wide range of phytohormones, as well as genes associated with the ABA signaling network, were examined in this study. However, hormonal crosstalk plays a significant role, particularly in the plant response to flooding. The utility of this research could be expanded by measuring phytohormones associated with the ET and GA pathways. Although effects of water stress on kiwifruit physiology and growth have been studied previously [38,39,40,41], these should be measured concomitantly with phytohormones and gene expression in any future research. 

Finally, chemical manipulation of ABA signaling for the development of ABA-based agrichemicals has been demonstrated in *Arabidopsis* with varying degrees of success [17] and in a number of other plant species [3]. The potential to mitigate stress in commercial horticultural crops using hormone-based treatments may offer a less time-consuming alternative to breeding for plant stress management. However, this requires a thorough understanding of the stress response process since hormone manipulation can also impact on yield.

Although there are many avenues for future research, this study has identified some useful biochemical and genetic markers for water stress studies in kiwifruit, which have provided novel information for this crop and may be pertinent to other water studies on perennial fruit crops. 

## 4. Materials and Methods

### 4.1. Plant Material

Four experiments (two each in 2020 and 2021) were carried out on clonal plants of two kiwifruit cultivars—*Actinidia chinensis* var. *chinensis* ‘Zesy002’ and *Actinidia chinensis* var. *deliciosa* ‘Hayward’. Table 2 shows the rationale for each of the experiments. Tissue cultured plantlets were transplanted individually from sealed plastic tubs containing an agar growth medium to 0.5 L planter bags filled to two thirds with Daltons™ GB mix (Daltons, Matamata, New Zealand) and topped up with a 50:50 ratio of potting mix and perlite. For the first two weeks of growth, the plantlets were put in a glasshouse under clear plastic tents to increase humidity and supplementary heating was applied. Following this the plantlets were grown under normal glasshouse conditions (15–24 °C, 14 h day length) to approximately 30 cm tall, with at least four fully expanded leaves. The plants were then transferred to 1.5 L pots filled with Daltons GB mix about 10 days before the experiment. A flood and drain table was used to water the plants for 12 min twice daily.

### 4.2. Watering Conditions 

Three watering treatments were established at time zero. Plants in the control treatment maintained the normal watering conditions from the flood and drain table with water to quarter pot height for 12 min twice a day. The drought treatment plants were placed on inverted trays above the water level of the flood and drain table and received no further water for the duration of the experiment. The waterlogged treatment plants were placed in solid trays filled with water and were subject to constant water to quarter pot height throughout the experiment. There were five replicates per treatment with each replicate comprising one plant to be destructively harvested per sample time point. The plants were arranged in a completely randomised block design. Total pot weight was measured at each of the times given in Table 2. 

### 4.3. Tissue Sampling for Phytohormone and Gene Expression Analysis

Leaf and root tissue samples were taken at times shown in Table 2. A preliminary trial had indicated that the kiwifruit plants began to wilt 48–72 h after withdrawal of water (also known as the ‘wilt point’) and this influenced the selection of the sampling times.

A portion of the youngest fully expanded leaf was sampled, removing major veins, to give enough tissue for analyses. Approximately 2–3 cm of root tissue was taken from the root tips and washed to remove soil. All tissue was snap frozen in liquid nitrogen and then stored at −80 °C until extraction and analysis. There were five replicates (individual plants) per time point for each treatment. Tissue samples were ground into a fine powder using a mortar and pestle with liquid nitrogen.

### 4.4. Phytohormone Analysis

#### 4.4.1. Reagents

Formic acid (Riedel-de Haën), jasmonic acid (JA), salicylic acid (SA), abscisic acid (ABA), methyl jasmonate (MeJA) and 2,5-dihydroxybenzoic acid (2,5-DHBA) were purchased from Sigma Aldrich (Auckland, New Zealand). Optima LC/MS grade acetonitrile and trifluoroacetic acid (TFA) were purchased from Thermo Fisher Scientific (Auckland, New Zealand). Jasmonoyl-isoleucine (JA-Ile), 12-hydroxyjasmonic acid (12-OH JA), 9,10-dihydrojasmonic acid (DHJA), cis-(+)-12-oxo-phytodienoic acid (cis-OPDA), (+/−)-4-(3-oxo-2-(pent-2-enyl)cyclopentyl) butanoic acid (OPC4) and indole-3-acetic acid (IAA) were purchased from Olchemim Ltd (Olomouc, Czech Republic). Phaseic acid (PA), dihydrophaseic acid (DPA), 7’-hydroxyabscisic acid (7’-OH-ABA) and abscisic acid D-glucopyranosyl ester (ABA-GE) were purchased from the National Research Council Canada (Saskatoon, SK, Canada). [^2^H_5_] JA, [^2^H_4_] SA and [^2^H_5_] MeJA were purchased from CDN Isotopes (Pointe-Claire, QC, Canada), [^2^H_6_] ABA and [^2^H_3_] 3,4-DHBA were purchased from Toronto Research Chemicals (Toronto, Ontario, Canada) and [^13^C_6_] IAA was purchased from Cambridge Isotopes (Andover, MA, USA). Salicylic acid O-β-glycoside (SAG) was synthesised following published methodology [42], and was >99% pure by 1H- and 13C-NMR. [^2^H_10_] JA-Ile was synthesised using a modification to published methodology by utilizing [^2^H_10_] L-isoleucine instead of isoleucine [43], and was >99% pure by liquid chromatography mass spectrometry (LC-MS). [^2^H_4_] SAG was synthesised similarly to SAG by utilizing [^2^H_4_] MeSA instead of MeSA as starting material and was >99% pure by LC-MS. Water was of Milli-Q grade.

#### 4.4.2. Phytohormone Extraction 

Samples of ground tissue were weighed (100 mg FW) and to each was added chilled (4 °C) extraction solvent (acetonitrile + 0.01% TFA) (1mL), labelled internal standard mix (2.5 ng [^2^H_4_] SA, 25 ng [^2^H_5_] JA, 6.4 ng [^2^H_6_] ABA, 0.6 ng [^2^H_10_] JA-Ile, 5 ng [^2^H_4_] SAG, 6.25 ng [^2^H_3_] 3,4-DHBA, 25 ng [^13^C_6_] IAA and 10 ng [^2^H_5_] MeJA) and stainless steel beads 0.9–2 mm (0.8 g) (Next Advance Inc., Raymertown, NY, USA). Samples were bead beaten for 5 min (Bullet Blender 24 Gold, Next Advance Inc., NY, USA) before being extracted overnight at 4 °C using an end-over-end rotator at 30 rotations/min. After centrifugation at 16,000× *g* for 5 min, supernatant from each sample was transferred into a well in a 96-well collection plate (Phenomenex, Torrance, CA, USA). The remaining pellet was re-extracted with the chilled (4 °C) extraction solvent (0.5 mL), combined with the first supernatant, and evaporated to dryness using a CentriVap concentrator (Labconco, Kansas City, MO, USA) at −4 °C. Sample clean-up employed graphitized carbon following a method described by Cai et al. [44], with modifications to adapt to a 96-well plate format. Briefly, samples were reconstituted in chilled (4 °C) 80:20 acetonitrile:water (1 mL) and shaken for 20 min on a flat-bed orbital shaker before SPE clean-up on a Hypersep Hypercarb 96-well plate (25 mg/1 mL; Thermo Scientific, CA, USA). Plates were conditioned using acetonitrile (1 mL) followed by water (1 mL). After conditioning, samples were loaded and the acidic plant hormones were eluted with acetonitrile (0.5 mL) and evaporated to dryness using a CentriVap concentrator (Labconco, Kansas City, MO, USA) at −4 °C. Samples were reconstituted in 10:90 acetonitrile:water (0.2 mL) for analysis by LC-MS. 

#### 4.4.3. LC-MS Analysis 

LC-MS/MS experiments were performed on a 5500 QTrap triple quadrupole/linear ion trap (QqLIT) mass spectrometer equipped with a TurboIon-Spray™ interface (AB Sciex, ON, Canada) coupled to a Shimadzu Exion UHPLC (Kyoto, Japan). Phytohormones were separated on a Poroshell 120 SB-C18 2.7 μm 2.1 × 150 mm ID column (Agilent Technologies, CA, USA) at 60 °C. Solvents were (A) water + 0.1% formic acid and (B) acetonitrile + 0.1% formic acid. The flow rate was 0.6 mL/min. The initial mobile phase of 2% B was held for 3 min, before ramping linearly to 16% B at 3.5 min, then to 100% B at 7 min and holding at 100% B for 1 min before resetting to the original conditions. The injection size was 10 μL. MS data were acquired in the negative mode, and positive mode (IAA and MeJA), using an MRM method with optimised Q1 and Q3 transitions for each analysed acidic phytohormone (Appendix A). Other operating parameters were as follows: dwell time, 10 ms; ionspray voltage, −4500 V; ionspray voltage (IAA and MeJA), 4500 V; temperature, 600 °C; curtain gas, 45 psi; ion source gas 1, 60 psi; ion source gas 2, 60 psi. Data were analysed using Analyst version 1.7.2 and SciexOS version 2.0 software packages. Concentrations were calculated based on the peak area for the endogenous compounds relative to those determined for the internal standards.

### 4.5. Gene Expression Analysis

#### 4.5.1. Sample Preparation/RNA Extraction

100 mg of ground tissue was weighed out and total RNA was extracted using the Spectrum Plant Total RNA kit (Sigma-Aldrich, Auckland, New Zealand) following the supplier’s recommendations. A Nanodrop 200c spectrophotometer (Thermo Scientific, Waltham, MA, USA) was used to determine the RNA concentration and purity. 

The RNA samples were sent on dry ice to Grafton Clinical Genomics (the University of Auckland, Auckland, New Zealand) where the NanoString analysis was carried out. The RNA concentrations and quality were checked using a PipeJet R Nanodispenser (BioFluidix GmbH, Freiburg, Germany) before working aliquots of 50 ng/µL were prepared with nuclease-free water and stored at −80 °C. 

#### 4.5.2. Gene Selection 

Reference genes (RGs), selected based on previous studies [45,46,47] and following preliminary trials to establish the most stably expressed RGs under the specific conditions of each experiment were: *Actin2*, *Glyceraldehyde 3-phosphate dehydrogenase* (*GAPDH*), *Protein phosphatase 2A* (*PP2A*) and *ubiquitin-conjugating enzyme* (*UBC*, used in GFS3 and GFS4 only) (Table 3).

Genes of interest (GoIs) were chosen based on their involvement in water stress and/or aspects of the ABA regulatory network, as shown by other published studies, or because they were reliable markers of other phytohormonal pathways (Table 3). We identified the most likely kiwifruit orthologues of genes previously identified in the literature by a combination of searches of the EnsemblPlants [48,49] and GenomicusPlants [50] databases using the relevant query gene. This process allowed us to identify the kiwifruit gene clades closest to the query gene (EnsemblPlants), efficiently check their likely identity by descent relationship by looking for evidence of micro-synteny between the genomic region containing the original query gene and micro-syntenic regions in the kiwifruit genome and confirm the two processes identified the same kiwifruit gene(s). A more detailed description of the gene mining methodology is given in Appendix A.

Oligomer design for the RG and GoI probes was carried out by NanoString Technologies Inc. (Seattle, WA, USA). The oligomer probes were synthesised by Integrated DNA Technologies Private Limited (IDT, Singapore). 

#### 4.5.3. Titration Analysis to Determine the Optimal RNA Input (ng) 

The optimum RNA input for PlexSet^®^ was determined by titration analysis as described previously [9], following 19 h hybridisation at 67 °C. The resulting optimal RNA input (185 ng per test sample for experiments 1 and 2, and 280 ng for experiments 3 and 4) was used in the full PlexSet runs. 

#### 4.5.4. Measurement of Gene Expression by PlexSet NanoString

Gene expression was measured by PlexSet^®^ NanoString, which provides direct counts of the number of molecules of interest expressed within a sample.

To enable hybridisation of the target RNA with the gene probes and PlexSet tags (24-Plex), a 15 µL reaction volume was set up for each sample containing: 0.5 µL each of Probe A (0.6 nM) and Probe B (3 nM) (mixed probe pools supplied by IDT and pre-diluted in TE-Tween buffer, pH 8.0 (10 mM Tris-HCl, 1mM disodium EDTA, 0.1% *v*/*v* Tween 20)); 5 µL NanoString hybridization buffer; 2 µL of NanoString PlexSet tags (Catalog number 121200002); 100 ng of sample RNA and water to bring the final volume to 15 µL. Samples were hybridised in a thermocycler (Mastercycler^®^ nexus, Eppendorf, Hamburg, Germany) at 67 °C for 19 h. They were then processed on a nCounter^®^ Max platform (NanoString Technologies Inc., Seattle, WA, USA) according to manufacturer instructions. This involved a two-step process where samples are first purified and immobilized onto a cartridge, followed by digital gene counting.

#### 4.5.5. Data Analysis 

The count of hybridised target sequences for each gene, generated from PlexSet, were analysed in nSolver™ (version 4.0) provided by NanoString Technologies Inc. (Seattle, WA, USA). It was ensured that the quality control parameters relating to image capture and reporter probe binding density were met and the data were then normalised against the RGs. Normalised data were then statistically analysed. 

**Table 3 ijms-24-07580-t003:** Genes of interest for analysis of water stress responsiveness, with measurement by PlexSet^®^ NanoString.

Identification (NCBI Entry/ Acc#/Achn#)	Gene Name (Abbreviation)	Reference for Previously Identified Kiwifruit Genes	Reference for Kiwifruit Genes Identified in This Analysis (Published Orthologue ^4^)
FG499230/ Acc06864.1/Achn146991	Pathogenesis-related protein family 1 (PR1)	[45]	-
Acc10322.1	Zeaxanthin epoxidase (ABA1/ZEP)	[47]	-
Acc06947.1	ABA-responsive element binding factor 4 (ABF4)	-	[51]
Acc03929.1	β-1,3-Glucanase 1/2 (β-glucanase)	[52]	-
Acc01538.1 ^1^	Dehydration-responsive element-binding protein 2 (DREB2)	-	[23]
Acc33422.1	Ethylene receptor 1 (ETR1)	-	[53]
Acc21754.1	9-cis-epoxycarotenoid dioxygenase 3 (NCED3)	-	[12]
Acc03466.1	Responsive to desiccation 29B (RD29B)	-	[54]
Acc291141.1	Responsive to desiccation 22 (RD22)	[47]	-
Acc28747.1 ^2^	Myelocytomatosis 2-like (MYC2-like)	[55]	-
Acc17926.1	W-tryptophan, R-arginine, K-lysine, Y-tyrosine 40 (WRKY 40)	-	[23]
Acc22643.1	Cytochrome P450, family 707, subfamily A, polypeptide i (CYP707A-i)	-	[12]
Acc28692.1	Cytochrome P450, family 707, subfamily A, polypeptide ii (CYP707A-ii)	-	[12]
Acc29493.1	Cytochrome P450, family 707, subfamily A, polypeptide iii (CYP707A-iii)	-	[12]
Acc02823.1	Abscisic acid insensitive 4i (ABI4-i)	-	[12]
Acc01553.1	Abscisic acid insensitive 4i (ABI4-ii)	-	[12]
Acc21776.1 ^3^	UDP-glycosyltransferase 71K1 (UTC-71C5)	-	[19]
Acc15636.1	Respiratory burst oxidase homolog protein F (RBOHF)	[31]	-

^1^ Gene mining analyses with the *Arabidopsis* genes indicated that DREB2A and DREB2B are Brassicacea-specific clades, making it more difficult to identify the closest orthologs in species outside of the Brassicaceae. The kiwifruit candidate tested is actually closer to a DREC2C ortholog. ^2^ This is actually bHLH14—a very close relative of MYC2, which responded better than all the other MYC2 candidates tested in kiwifruit [55]. ^3^ The UGT71C family appears to have diverged very substantially because true orthologs by either genome position or tree position are difficult to find. The kiwifruit candidate tested is closest to the *Arabidopsis* UGT71C5 candidate by its tree position. ^4^ The published orthologs are from species other than kiwifruit.

### 4.6. Statistical Analysis 

For the gene expression data, analysis of variance (ANOVA) was used to compare expression levels. Data were log transformed before analysis. The treatments consisted of a factorial combination of water-stress and time, plus a time 0 unstressed sample; in the ANOVA this was handled by fitting a factor for time 0 vs. later, and then testing the water stress and time effects nested within that time 0 factor.

Where there were significant effects in the ANOVA, means were compared using Fisher’s least significant difference (*p* = 0.05, no adjustment for multiple testing). Means were back transformed from the log scale and expressed as fold change relative to the control treatment at the same time-point.

For the phytohormone data, the same ANOVA approach was used. Data were log transformed before analysis; concentrations below the limit of detection (LoD) were replaced with 0.5× LoD before taking logs.

All analysis was done using GenStat (version 20, VSNi Ltd., Hemel Hempstead, UK, 2020).

## 5. Conclusions

This research investigated the biochemical and molecular basis of the kiwifruit response to water stress (drought and waterlogging) by measuring marker genes across the ABA signalling network and stress-induced phytohormones. Of all the phytohormones measured, endogenous concentrations of ABA increased significantly in drought conditions, especially in the roots, and were associated with upregulation of the ABA synthesis gene *NCED3*. Conversely, decreased ABA concentrations in waterlogged roots correlated with increased activity of ABA catabolic genes (*CYP707As*). The study has identified molecular markers and shown that water stress extremes induced strong phytohormone/ABA gene responses in the roots, which are the main site of water stress perception. Understanding responses and how to manipulate them will help to increase kiwifruit resilience and productivity during water stress conditions, which is becoming more crucial as the pace of climate change accelerates. 

## Figures and Tables

**Figure 1 ijms-24-07580-f001:**
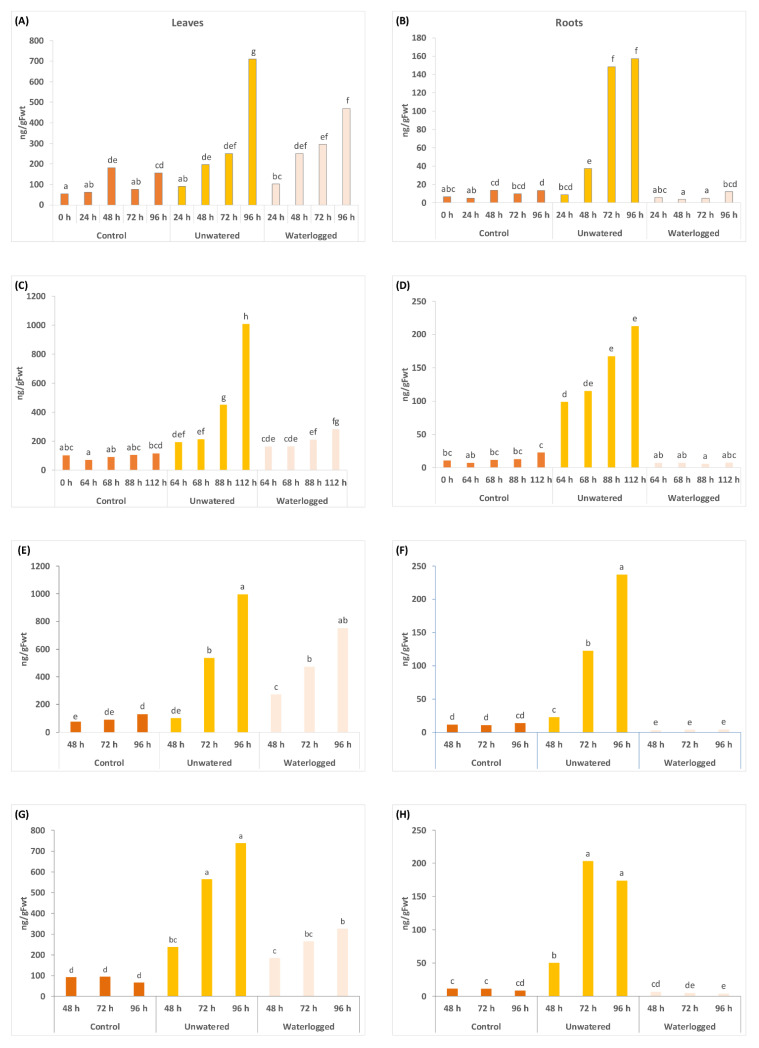
Abscisic acid (ABA) concentrations measured in leaves (left panel) and roots (right panel) at various sample times (in hours, shown in the x-axes) after application of three treatments. From time 0 h, plants in the unwatered treatment were not watered and waterlogged plants were subjected to constant water at quarter pot height. Plants in the control treatment received the standard watering regime of 12 min, twice daily. There were five replicate plants/time/treatment. Two separate experiments were carried out in 2020 on *Actinidia chinensis* var. *chinensis* ‘Zesy002’ plants over a 96 h treatment period as shown in (**A**,**B**), and 112 h (**C**,**D**). In 2021, another two experiments were carried out using an identical setup to compare responses between ‘Zesy002’ (**E**,**F**), and *A*. *chinensis* var. *deliciosa* ‘Hayward’ (**G**,**H**). Different lettering over the bars indicates statistically significant differences, as shown by Fisher’s Least Significant Difference (LSD), *p* ≤ 0.05, within each separate experiment and tissue type.

**Figure 2 ijms-24-07580-f002:**
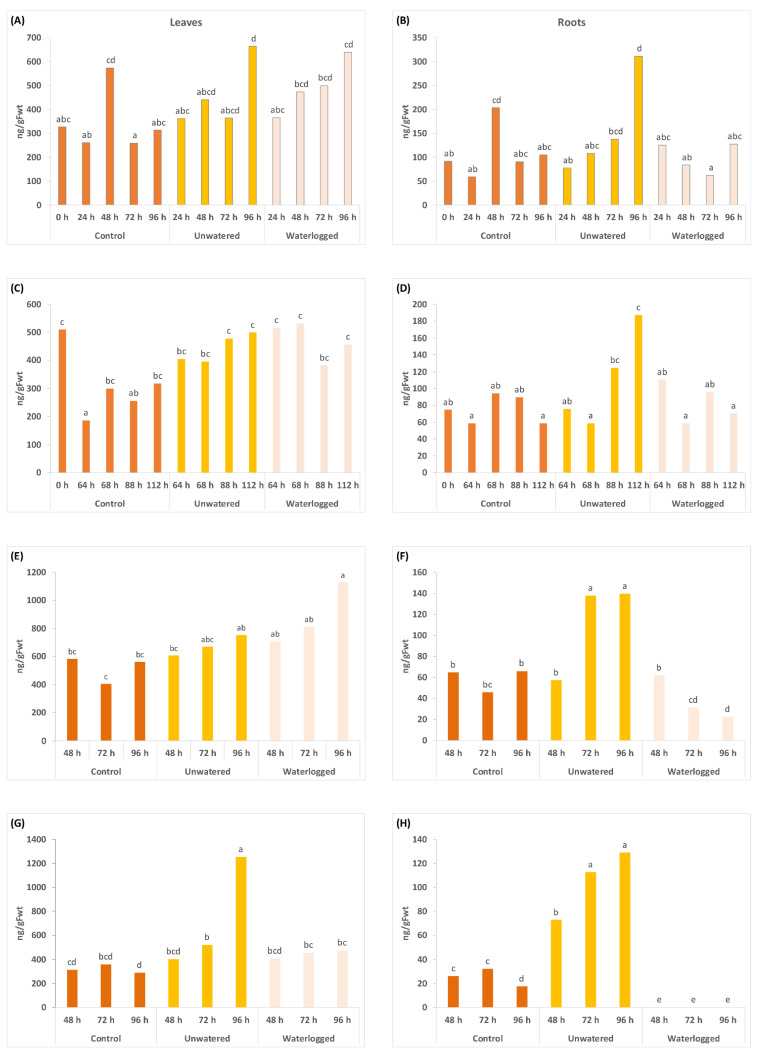
Dihydrophaseic acid (DPA) concentrations measured in leaves (left panel) and roots (right panel) at various sample times (in hours, shown in the x-axes) after application of three treatments. From time 0 h, plants in the unwatered treatment were not watered and waterlogged plants were subjected to constant water at quarter pot height. Plants in the control treatment received the standard watering regime of 12 min, twice daily. There were five replicate plants/time/treatment. Two separate experiments were carried out in 2020 on *Actinidia chinensis* var. *chinensis* ‘Zesy002’ plants over a 96 h treatment period as shown in (**A**,**B**), and 112 h (**C**,**D**). In 2021, another two experiments were carried out using an identical setup to compare responses between ‘Zesy002’ (**E**,**F**), and *A*. *chinensis* var. *deliciosa* ‘Hayward’ (**G**,**H**). Different lettering over the bars indicates statistically significant differences, as shown by LSD, *p* ≤ 0.05, within each separate experiment and tissue type.

**Figure 3 ijms-24-07580-f003:**
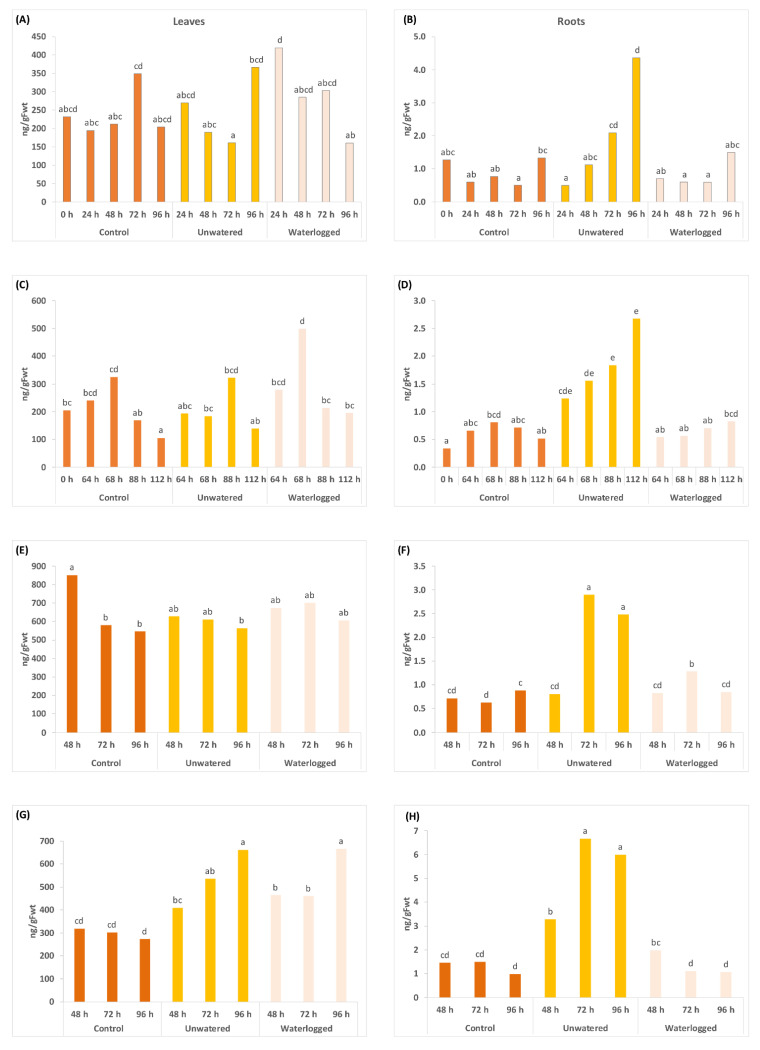
Phaseic acid (PA) concentrations measured in leaves (left panel) and roots (right panel) at various sample times (in hours, shown in the x-axes) after application of three treatments. From time 0 h, plants in the unwatered treatment were not watered and waterlogged plants were subjected to constant water at quarter pot height. Plants in the control treatment received the standard watering regime of 12 min, twice daily. There were five replicate plants/time/treatment. Two separate experiments were carried out in 2020 on *Actinidia chinensis* var. *chinensis* ‘Zesy002’ plants over a 96 h treatment period as shown in (**A**,**B**), and 112 h (**C**,**D**). In 2021, another two experiments were carried out using an identical setup to compare responses between ‘Zesy002’ (**E**,**F**), and *A*. *chinensis* var. *deliciosa* ‘Hayward’ (**G**,**H**). Different lettering over the bars indicates statistically significant differences, as shown by LSD, *p* ≤ 0.05, within each separate experiment and tissue type.

**Figure 4 ijms-24-07580-f004:**
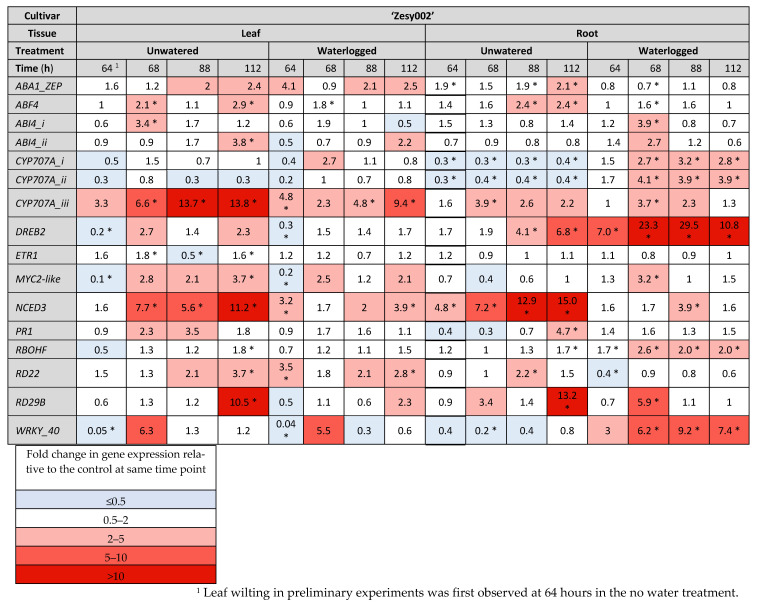
Fold-change in gene expression induced by water stress conditions in leaf and root tissue of *Actinidia chinensis* var. *chinensis* ‘Zesy002’ kiwifruit plants after 64 h, 68 h, 88 h and 112 h of water stress treatments in 2020. From time 0 h, plants in the unwatered treatment were not watered and waterlogged plants were subjected to constant water at quarter pot height. Plants in the control treatment received the standard watering regime of 12 min, twice daily. There were five replicate plants/time/treatment. Gene expression was quantified by PlexSet^®^ NanoString using *Actin2*, *Glyceraldehyde 3-phosphate dehydrogenase* (*GAPDH*) and *Protein phosphatase 2A* (*PP2A*) as the reference genes. The genes of interest are shown in the left-most column. Statistical analysis was performed at the log_2_ scale, and the heat map shows fold change data between water stressed plants and the control at the same time point. Fold-change data in red shades (defined in the color panel) indicate an up-regulation relative to the control, whilst blue shades represent down-regulation relative to the control. Fold changes of 0.5–2 are often considered to be statistically unreliable [25,26,27]. Statistically significant differences from the control at individual time points, as determined by ANOVA at (*p* ≤ 0.05) are identified with an asterisk.

**Figure 5 ijms-24-07580-f005:**
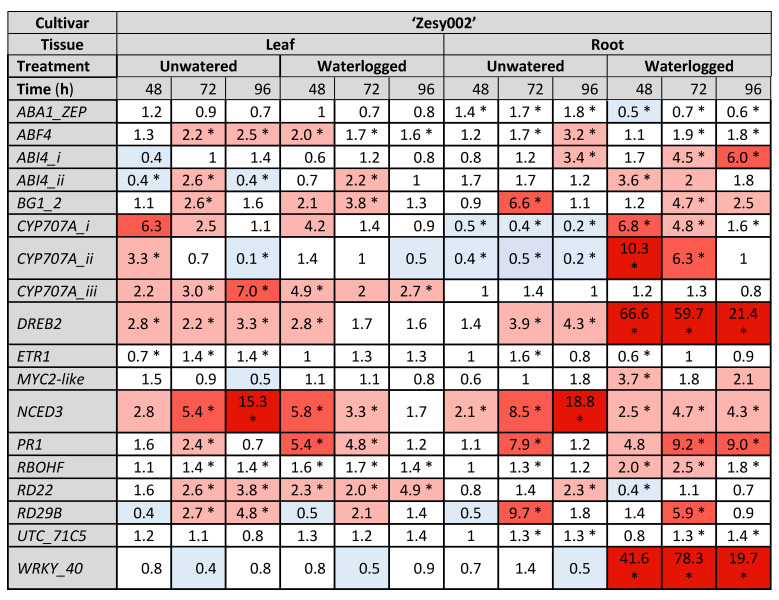
Fold-change in gene expression induced by water stress conditions in leaf and root tissue of *Actinidia chinensis* var. *chinensis* ‘Zesy002’ kiwifruit plants after 24 h, 48 h, 72 h and 96 h of water stress treatments in 2021. Plants in the unwatered treatment did not receive any watering from time 0 h, while waterlogged plants were subjected to constant water at quarter pot height. Plants in the control treatment received the standard watering regime of 12 min, twice daily. There were five replicate plants/time/treatment. Gene expression was quantified by PlexSet^®^ NanoString using *Actin2*, *Glyceraldehyde 3-phosphate dehydrogenase* (*GAPDH*), *Protein phosphatase 2A* (*PP2A*) and *ubiquitin-conjugating enzyme* (*UBC*) as the reference genes. The genes of interest are shown in the left-most column. Statistical analysis was performed at the log_2_ scale, and the heat map shows fold change data between water stressed plants and the control at the same time point. Red shades (as defined in Figure 2) indicate an up-regulation relative to the control, whilst blue represents down-regulation relative to the control. Fold changes of 0.5–2 are often considered to be statistically unreliable [25,26,27]. Statistically significant differences from the control at individual time points, as determined by ANOVA at (*p* ≤ 0.05) are identified with an asterisk.

**Figure 6 ijms-24-07580-f006:**
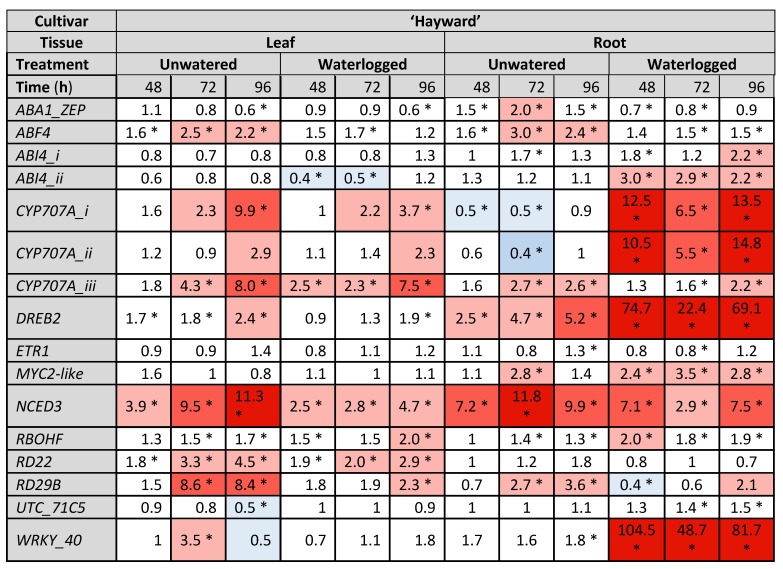
Fold-change in gene expression induced by water stress conditions in leaf and root tissue of *Actinidia chinensis* var. *deliciosa* ‘Hayward’ kiwifruit plants after 24 h, 48 h, 72 h and 96 h of water stress treatments in 2021. Plants in the no water treatment did not receive any watering from time 0 h, while waterlogged plants were subjected to constant water at quarter pot height. Plants in the control treatment received the standard watering regime of 12 min, twice daily. There were five replicate plants/time/treatment. Gene expression was quantified by PlexSet^®^ NanoString using *Actin2*, *Glyceraldehyde 3-phosphate dehydrogenase* (*GAPDH*), *Protein phosphatase 2A* (*PP2A*) and *ubiquitin-conjugating enzyme* (*UBC*) as the reference genes. The genes of interest are shown in the left-most column. Statistical analysis was performed at the log_2_ scale, and the heat map shows fold change data between water stressed plants and the control at the same time point. Red shades (as defined in Figure 2) indicate an up-regulation relative to the control, whilst blue represents down-regulation relative to the control. Fold changes of 0.5–2 are often considered to be statistically unreliable [25,26,27]. Statistically significant differences from the control at individual time points, as determined by ANOVA at (*p* ≤ 0.05) are identified with an asterisk.

**Table 1 ijms-24-07580-t001:** Relative weight (%) of unwatered and waterlogged pots compared with the control over time for 4 experiments carried out on *A. chinensis* var *chinensis* ‘Zesy002’ (Experiments 1–3) and *Actinidia chinensis* var *deliciosa* ‘Hayward’ (Experiment 4) clonal kiwifruit plants. Probabilities (*p*-values) from ANOVA refer to within-experiment comparisons.

Experiment	Treatment	Time (h)			
		24	48	72	96
1	Unwatered	93 ± 2	83 ± 1.6	76 ± 2	72 ± 2.3
	Waterlogged	108 ± 2	105 ± 1.8	110 ± 2.3	111 ± 2.8
	*p*-value	<0.001	<0.001	<0.001	<0.001
2		64	68	88	112
	Unwatered	76 ± 1.6	74 ± 1.6	69 ± 1.5	70 ± 1.93
	Waterlogged	109 ± 1.9	109 ± 1.9	107 ± 1.8	117 ± 2.3
	*p*-value	<0.001	<0.001	<0.001	<0.001
3		24	48	72	96
	Unwatered	91 ± 1.5	84 ± 1.6	76 ± 1.4	70 ± 1
	Waterlogged	110 ± 1.6	114 ± 1.9	116 ± 1.6	113 ± 1.2
	*p*-value	<0.001	<0.001	<0.001	<0.001
4		24	48	72	96
	Unwatered	91 ± 2.3	81 ± 2.5	75 ± 2.8	70 ± 2.1
	Waterlogged	112 ± 2.5	109 ± 2.9	111 ± 3.3	111 ± 2.5
	*p*-value	<0.001	<0.001	<0.001	<0.001

**Table 2 ijms-24-07580-t002:** Sampling times and rationale for the four water stress experiments conducted on *Actinidia chinensis* var *deliciosa* ‘Hayward’ and *A. chinensis* var *chinensis* ‘Zesy002’ clonal kiwifruit plants.

Cultivar	Experiment Start Date (Experiment #)	Sampling Times after Experiment Start (h)	Experimental Rationale
‘Zesy002’	30 November 2020 (# 1)	24, 48, 72 and 96	Pilot study to investigate hormone levels under water stress (and to help guide gene selection).
‘Zesy002’	11 December 2020 (# 2)	64 ^1^, 68, 88 and 112	The experiment was carried out over a longer period to induce more severe water stress and included both phytohormone and gene measurements.
‘Zesy002’	15 March 2021 (# 3)	48, 72 and 96	Repeat of exp 1 trial setup (with sample times before and after plants appear visibly water stressed) but with full phytohormone and gene measurements.
‘Hayward’	21 March 2021 (# 4)	48, 72 and 96	Identical design to exp 3, but in a different cultivar.

^1^ Point at which the kiwifruit plants start to show visible signs of water stress, i.e., wilting occurs after withdrawal of water, as determined by a preliminary experiment.

## Data Availability

Data are contained within the article or the Appendix A.

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
