# Peer review of "Effects of Drought and Flooding on Phytohormones and Abscisic Acid Gene Expression in Kiwifruit"

_ijms, 2023, doi:10.3390/ijms24087580_

Round 1

Reviewer 1 Report

Involvement of phytohormones, especially ABA, in drought stress is well-known and many reports address this point and the molecular processes behind it. However, most of these reports are based on work with model plants, mostly Arabidopsis. 

The novelty of the present report is, that it investigates Kiwi-response and that it compares drought-response with flooding (which is not well understood). In addition, both, leaves and root responses ) are compared. Therefore, this report shows valuable new and interesting data worth to be published.

In general, the data are convincing and no major changes have to be done. However, there are some minor points:

Line 152: should read "(left hand panel)"

All Figures show separately results from four different experiments (different cultivars, different years). I was wondering if this could be somehow combined, e.g. results from two years combined in one graph. But I also know that this is difficult and then the chosen way of presentation is fine.

The authors investigated ABA-content and expression of ABA-related genes in both, leaves and roots. In discussion they state that responses (gene expression) in roots is more pronounced. However, ABA content is much higher in eaves. This somehow discrepancy should be explained.

In general, DISCUSSION in my opinion should be shortened, more focussed and more concise. It is very general, sometimes written like a textbook on ABA. 

Reviewer 2 Report

This research investigated the biochemical and molecular basis of the kiwifruit response to water stress (drought and waterlogging) by measuring marker genes across the ABA signalling network and stress-induced phytohormones. Understanding responses and how to manipulate them will help to increase kiwifruit resilience and productivity during water stress conditions, which is becoming more crucial as the pace of climate change accelerates. Overall, the authors have done massive work evaluating the impact of drought and waterlogging stresses on the expression of marker genes across hormone pathways. I have only a fewminor suggestions for further corrections

Please use some diverse keywords. Keywords should not have been used in the title.

Line 34-44, merge these two short paragraphs and replace an older reference with a recent study about drought stress such as doi: 10.1002/tpg2.20279. This reference can also be cited at suitable place in the discussion.

Line 570, why these two cultivars were selected? two kiwifruit cultivars – Actinidia chinensis var. chinensis ‘Zesy002’ and Actinidia chinensis var. deliciosa ‘Hayward’.

Minor editing of English language required
